# ELICITING HUMAN-LIKE SOCIAL REASONING IN LARGE LANGUAGE MODELS

## ABSTRACT

Large language models (LLMs) have gained significant attention for their potential to replicate human participants in social science simulations. However, previous works on LLM reasoning focus on enhancing the capabilities for math and logical problems, overlooking the reasoning process behind social behavior, such as controversial social attitudes, moral dilemmas, and economic games. In this study, we explore the limitations of current models and propose a new approach to improve their human-likeness in social behavioral reasoning tasks. We introduce the Social-Behavioral-Reasoning (SBR) dataset, comprising 1,560 quadruples of human profiles, social questions, reasoning processes, and final choices. Utilizing this dataset, we evaluate large reasoning models (LRMs), revealing a contradiction: while LRMs increase society-level diversity, they fail to maintain individual-level accuracy. Our findings further indicate that the observed increase in diversity is primarily attributed to random variation introduced by longer reasoning durations, rather than improved understanding of human diversity. To address these issues, we propose the Reasoning-Enhanced-SFT method, which explicitly aligns both the reasoning and final choices with human data. Our experimental results demonstrate that our method significantly improves both in-domain and out-of-domain performance, enhancing the generalization ability across diverse social contexts. Our user study results confirm the model's ability to produce a reasoning process more closely aligned with specific human reasoning patterns. Our work offers a new pathway to overcome the challenges that limit the use of LLMs in social simulations. Aligning model outputs with human reasoning boosts LLMs' credibility and applicability in social science, enabling more precise and insightful simulations of human behavior.

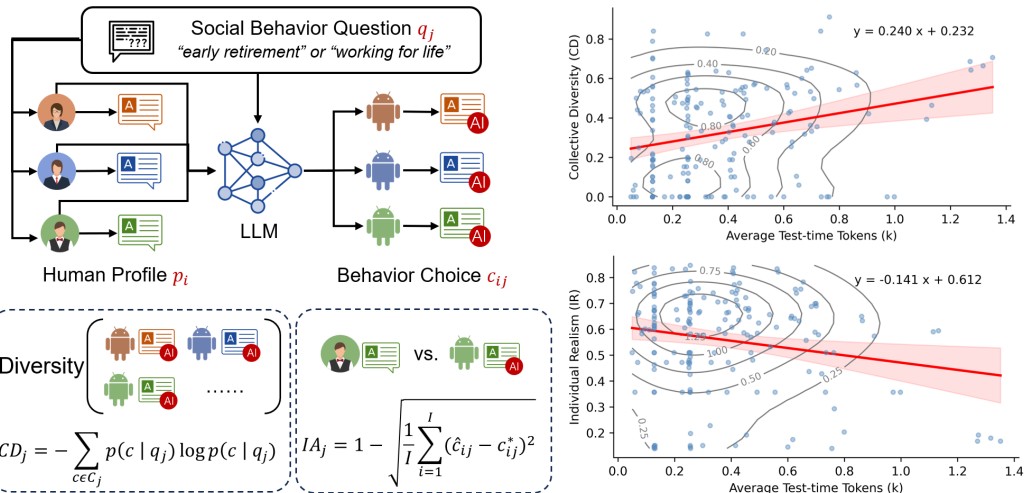

Figure 1: **Current Large Language Models' Test-Time Reasoning Fails in Human Social Behaviors Simulation.** We examines LLMs' ability to simulate human social behaviors in response to questions $q_j$ using profiles $p_i$ to choices $c_{ij}$. We assess the simulation's quality by two metrics: Collective Diversity(CD) and Individual Realism (IR). The right graphs show that more test-time tokens boost diversity but reduce realism, showing a diversity-realism trade-off.

# 1 INTRODUCTION

Large language models (LLMs) have proven to be valuable tools in social science simulation, offering extensive scalability and novel research opportunities Farrell et al. (2025); Park et al. (2023); Gao et al. (2024); Zeng et al. (2025). However, challenges arise when LLMs attempt to replicate human participants in simulation, including producing flat and stereotypical responses, failing to capture the complexity and diversity of human behavior, and misrepresenting specific demographic groups Wang et al. (2025a;b); Bisbee et al. (2024); Santurkar et al. (2023); Murthy et al. (2024). Inspired by Behavioral Reasoning Theory (BRT) proposed by James D. Westaby Westaby (2005) which emphasizes the importance of underlying reasoning processes that lead to human actions, aligning LLMs with social behavioral reasoning, rather than merely with the behaviors, offers a promising avenue for addressing these challenges, especially in complex and controversial social contexts. Although recent large reasoning models (LRMs) epitomized by Deepseek-r1 and OpenaiO1 Guo et al. (2025); Jaech et al. (2024) have acquired impressive reasoning ability on formal tasks grounded in logic, mathematics, and code, their capacity for social behavioral reasoning remains unvalidated, which involves subjective, context-dependent judgments influenced by individual profiles.

To bridge this gap, we first introduce the **Social-Behavioral-Reasoning** (SBR) dataset, providing a solid foundation for evaluating LRMs' social behavioral reasoning ability. The SBR dataset covers diverse social topics, containing 1,560 high-quality quadruples consisting of human profile, social question, reasoning process, and final choice in a unified structure. The SBR dataset is the first open-source resource to simultaneously supply fine-grained individual profiles and transparent detailed social behavioral reasoning process.

Based on this dataset, we conduct a comprehensive evaluation of LRMs' ability in social behavioral reasoning. Our evaluation mainly focuses on how LRMs' test-time reasoning influences their performance in social behavioral reasoning tasks from the perspectives of collective diversity and individual realism. We find that LRMs achieve higher collective diversity, they fail to maintain individual realism. Moreover, the observed increase in diversity arises primarily from stochastic branching during extended reasoning, rather than from genuine improvements in the models' ability to capture diverse human perspectives.

To address the limitations of current LRMs, we introduce **Reasoning-Enhanced-SFT** method, which incorporates the reasoning process as a crucial component of the training data, marking a significant step forward from previous training approaches that focused solely on output behavior. This design not only encourages the model to learn how humans take action but also how humans reason in complex and controversial situations, thereby improving generalization across unseen social contexts.

Our experiments demonstrate that the Large-Human-Behavior-Reasoning-Model (LHBRM), trained using this method, achieves superior performance on both the test set of the SBR dataset and an out-of-domain moral dilemma dataset. It outperforms all comparable LRMs and surpasses models specifically trained on human behavior datasets. Beyond merely replicating human-like social behavior, we are also concerned with whether the model can generate human-like reasoning processes. To this end, we conducted a user study where participants were asked to select the reasoning process that most closely aligns with the human gold standard from three alternatives. Our LHBRM model achieved the highest selection rate across all profiles and questions, indicating its ability to produce reasoning processes that are closely aligned with human reasoning patterns.

The key contributions of this work are as follows:

- We construct the **Social-Behavioral-Reasoning** dataset that provides explicit ground truth for behavioral reasoning.

- We evaluate LRMs in social reasoning and reveal critical insights about the limitations of current test-time reasoning approaches.

- We propose the **Reasoning-Enhanced-SFT** method, which incorporates reasoning processes into supervised fine-tuning, enabling models to learn the underlying reasoning mechanisms.

- We demonstrate through experiments and a user study that our method significantly improves both in-domain and out-of-domain performance, and generates reasoning processes that are perceived as more human-like.

## 2 RELATED WORKS

### 2.1 LLMS REPLACE HUMAN PARTICIPANTS IN SOCIAL SCIENCE

Recent studies explore using LLMs as replication for human participants in social science experiments and agent-based simulations Li et al. (2024); Chen et al. (2023); Gao et al. (2023); Xie et al. (2024); Hayati et al. (2023); Zeng et al. (2025). While this approach offers scalability and novel experimental possibilities, it also raises concerns: models tend to produce flattened or stereotyped responses, misrepresent minority or marginalized groups, and lack psychological consistency or profile specificity in reasoning across repeated trials Wang et al. (2025a); Bisbee et al. (2024); Santurkar et al. (2023). Existing fixes,such as prompt engineering Wang et al. (2025a); Dimgba et al. (2025), or supervised finetuning Murthy et al. (2024), largely focus on matching superficial or outcome level behavior rather than aligning the underlying reasoning process that leads to decisions, leaving the challenge of building faithful and consistent LLM participants unresolved.

### 2.2 LARGE REASONING MODEL

LRMs, exemplified by OpenAI O1 and DeepSeek R1 Guo et al. (2025); Jaech et al. (2024), leverage extended test-time reasoning methods such as chain-of-thought, self-consistency, and other inference-time search to improve performance on tasks with clear, objective correct answers Snell et al. (2024); Wang et al. (2023); Wei et al. (2022). Though these techniques yield gains in mathematical, logical, and formal reasoning benchmarks, reasoning needed in social science contexts is different: judgments are subjective, context-dependent, and shaped by individual profilesShao et al. (2024b). Thus, it remains unclear whether LRMs' reasoning methods actually produce reasoning aligned with human social reasoning rather than just improving correctness on formal tasks.

## 3 PRELIMINARY

### 3.1 PROBLEM STATEMENT

We investigate the human-likeness of LLMs and LRMs in social behavioral reasoning tasks. Specifically, we construct a set of social questions $Q = \{q_1, q_2, \ldots, q_J\}$,covering classical, complex, and controversial issues, and a set of human profiles $P = \{p_1, p_2, \ldots, p_I\}$. Given a profile $p_i \in P$ and a question $q_j \in Q$, the model $M_\theta$ (an LLM or LRM with parameters $\theta$) is prompted to generate: a reasoning process $r_{ij} = M_\theta^{(r)}(p_i, q_j)$, and a final behavioral choice $c_{ij} = M_\theta^{(c)}(p_i, q_j) \in \mathcal{C}_j$. Formally, the model defines a conditional distribution

$$M_\theta(c, r \mid p_i, q_j),$$

from which we obtain the reasoning process $r_{ij}$ and the final choice $c_{ij}$. Our interest is the degree to which $\{c_{ij}\}$ resembles the behavioral patterns of human responses under the corresponding profile and question.

### 3.2 EVALUATION METRICS

To quantitatively measure the human-likeness of model behaviors, we design two levels of metrics addressing two critical issues identified in prior workWang et al. (2025a): *flattening* (low diversity, ignoring minority profiles) and *misrepresentation* (failure to accurately role-play specific groups or individuals).

- **Collective Diversity (CD).** To address the flattening problem, we compute the entropy of model choices across all profiles for each question. For question $q_j$, let $p(c \mid q_j)$ be the empirical distribution of $\{c_{ij}\}_{i=1}^{I}$. The entropy is

$$H(q_j) = -\sum_{c \in \mathcal{C}_j} p(c \mid q_j) \log p(c \mid q_j).$$

The Society Diversity metric is the average entropy across all questions:

$$SD = \frac{1}{J} \sum_{j=1}^{J} H(q_j).$$

- **Individual Realism (IR).** To assess individual-level fidelity, we map the ordered choice list $\mathcal{C}_j$ of question $q_j$ onto the interval $[0, 1]$. Let $c_{ij}^* \in [0, 1]$ denote the true human choice of profile $p_i$ and $\hat{c}_{ij} \in [0, 1]$ the model-predicted choice. The Individual Accuracy (IA) is defined as 1-RMSE, where RMSE denotes the root mean squared error over all profiles and questions:

$$IA = 1 - \sqrt{\frac{1}{IJ} \sum_{i=1}^{I} \sum_{j=1}^{J} \left( \hat{c}_{ij} - c_{ij}^* \right)^2}.$$

## 4 DATASET

### 4.1 HUMAN-BEHAVIORAL-REASONING DATASET

To study whether LRMs can reproduce human thinking when people face social issues without a unique correct answer, we require data that couples human profile $p_i$, social question $q_j$, explicit reasoning process $r_{ij}$, and final choice $c_{ij}$. Existing open datasets do not satisfy this requirement. For example, social survey datasets such as American National Election Studies(ANES), The International Social Survey Programme(ISSP), and Moral Machine Awad et al. (2018) provide human profiles and final choices, but they lack explicit reasoning processes. Conversely, social content on platforms such as X, Reddit, or Weibo includes abundant expressions of opinion and reasoning, but lacks structured human profile information.

To fill this gap, we curate the **Social-Behavioral-Reasoning (SBR) dataset**, which provides high-quality quadruples $(p_i, q_j, r_{ij}, c_{ij})$. We select widely discussed topics in classical social science, construct a diverse set of questions, and design a questionnaire to elicit both explicit reasoning processes and final decisions. Figure 2 illustrates the dataset collection and data structure.

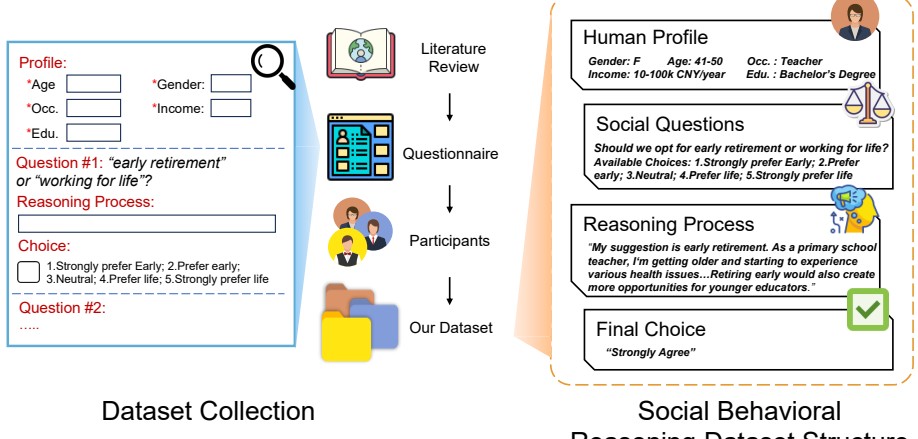

Figure 2: Overview of Social-Behavioral-Reasoning Dataset. Left: Survey design and collection protocol. Right: Dataset structure.

### 4.2 QUESTION DESIGN

We draw candidate questions from classical literature in social science Awad et al. (2018) and from prior studies that evaluate LLMs as a replication for human participants Ding et al. (2025); Santurkar

et al. (2023). To elicit meaningful reasoning, we make sure that questions require reflective thinking rather than purely factual recall. We also expect human responses to vary across individuals rather than tasks with unique correct answers. Finally, to reduce the understanding burden for participants, we prefer general, common, and direct questions rather than complex, professional questions. In total, we construct 15 social questions with corresponding choices list, which cover five broad topics: family attitude, society attitude, technology attitude, moral dilemma, and economic game. The complete question list is provided in Appendix A.1.

### 4.3 DATA COLLECTION

We develop an online survey to collect the quadruples and recruit participants from diverse demographic groups. We pay compensation for them above the local minimum wage. The questionnaire clearly states the purpose of the study and provides detailed instructions. To discourage careless answers, we implement two quality-control rules: (i) each question must take at least 90 seconds, and (ii) each reasoning process must exceed at least 50 words. After automatic checks, we manually review the submissions. We discard incomplete or low-quality responses. In the end, we obtain 104 valid questionnaires, each including answers to all 15 questions, which results in a total of 1,560 valid quadruples.

### 4.4 DATASET DESCRIPTION

Each entry in the dataset consists of four components. Figure 2 provides a schematic overview:

- **Human Profile** $p_i$**:** demographic features including gender (Male, Female), age (from 18 to 60), occupation(17 different occupation), education(from below junior high level to above graduate degree), and annual income (from below ¥10,000 to ¥500,000).
- **Social Question** $q_j$**:** 15 questions across five topics, each with an ordered choice list.
- **Reasoning Process** $r_{ij}$**:** free-text reasoning of at least 50 words for each item, the average length of reasoning process we collected is 74.39 words.
- **Final Choice** $c_{ij}$**:** the selected choice, normalized to a 0–1 scale according to the ordered list.

The online survey and SBR dataset is released at repository [*] after removing sensitive information. Descriptive statistics of demographic features and sample cases are reported in the appendix A.1.

## 5 EVALUATING THE BEHAVIORAL REASONING ABILITY OF LARGE REASONING MODELS

With the Soical-Behavioral-Reasoning dataset, we evaluate the human-likeness of LLMs and LRMs in social behavioral reasoning tasks. We focus on whether test-time reasoning enhances LRMs' human-like behavioral reasoning ability. Further more, we examine how different test-time reasoning budgets affect this reasoning ability.

### 5.1 Q1: DOES A LARGE REASONING MODEL POSSESS SOCIAL REASONING ABILITY?

#### 5.1.1 EVALUATION EXPERIMENT DESIGN

**Model Selection.** To study the effect of test-time reasoning on social behavioral reasoning, we filter ten comparison pairs that cover multiple families and sizes Each pair contrasts a model that uses test-time reasoning with a corresponding model that does not. We construct these pairs in two ways. First, we compare LRMs with its corresponding base LLMs, for example, DeepSeek-V3 versus DeepSeek-R1. Second, we compare the same LRM with test-time thinking enabled or disabled, for example Qwen3-plus with thinking mode on versus thinking mode off.

**Prompt Setting.** We adopt a role-playing prompt, providing a profile $p_i$ and a social question $q_j$. The model predicts a behavioral choice $c_{ij}$. If reasoning mode is enabled, the model also outputs an

---

[*]https://anonymous.4open.science/r/Human-like-Social-Reasoning-333B/

explicit reasoning process $r_{ij}$. All other decoding parameters, such as temperature and maximum length are kept identical. The full prompt format is shown in Appendix A.2.

### 5.1.2 RESULT

Table 1 reports the results. We observe three consistent patterns across different model families and sizes. First, LRMs produce higher society-level diversity than LLMs. This suggests that additional reasoning encourages exploration of minority choices and increases population level variation. Second, LRMs show mixed performance on group alignment. In some groups LRMs capture human group differences better than LLMs, while in other groups they miss important distinctions. In some model pairs they outperform LLMs, while in others they fail to capture meaningful distinctions. Third, at the individual level LRMs underperform relative to LLMs. In short, compared with LLMs, LRMs perform particularly well on society level metrics but perform particularly poorly on individual level metrics. In other words, **extra reasoning effort increases society-level diversity while it reduces fidelity at the individual level.**

| Series | DeepSeek | | | | | | Microsoft | | GLM | | | | Qwen | | | |
|---|---|---|---|---|---|---|---|---|---|---|---|---|---|---|---|---|
| Model Name | Deep-seek-V3 | Deep-seek-R1 | Llama 3.1-8B | Deep-seek-R1-8B | Llama 3.3-70B | Deep-seek-R1-70B | phi-4 | phi-4-reason | GLM-4-9B-0414 | GLM-Z1-9B | GLM-4.5 ReasonOff | GLM-4.5 | Qwen 2.5-32B | Qwen-QwQ-32B | Qwen 3-Plus ReasonOff | Qwen 3-Plus |
| CD ↑ | 0.317 | **0.364** | 0.214 | **0.524** | 0.088 | **0.414** | **0.222** | 0.040 | 0.210 | **0.373** | 0.268 | **0.355** | 0.116 | **0.304** | 0.149 | **0.297** |
| IR ↑ | **0.607** | 0.590 | **0.593** | 0.546 | **0.570** | 0.566 | **0.605** | 0.405 | **0.560** | 0.501 | **0.592** | 0.581 | **0.608** | 0.603 | **0.582** | 0.567 |

Table 1: Comparison of LLMs and LRMs on collective diversity(CD) and individual realism(IR). Bold numbers highlight the better score between LLM and LRM pair. If the LLM outperforms the LRM, the value is highlighted in **red**.

## 5.2 Q2: HOW DOES TEST-TIME REASONING INFLUENCE SOCIAL BEHAVIRAL REASONING PERFORMANCE?

Q1 result reveals a striking phenomenon. LRMs show opposite trends across metrics, performing well at the society level but introducing large errors at the individual level. This suggests that the observed increase in diversity may not reflect genuine representation of minority groups. Instead, it may result from random variation introduced by longer test-time reasoning. We hypothesize that higher diversity arises from stochastic branching during extended reasoning rather than improved modeling of underrepresented populations. To test this hypothesis, we control the thinking budget of LRMs and examine how the length of test-time reasoning affects social reasoning performance.

### 5.2.1 EVALUATION EXPERIMENT DESIGN

We select LRMs that allow explicit control of the thinking budget. We set three levels of reasoning tokens and run experiments under each setting. The prompt structure remains consistent with the previous section.

### 5.2.2 RESULT

Table 2 and Figure 1 report the results. We find that as the number of reasoning tokens increases, society-level diversity improves but individual accuracy decreases. This pattern supports our hypothesis:

Current LRMs trained mainly on mathematical, logical, or programming tasks **do not** effectively enhance social behavioral reasoning ability, especially human-likeness. While extended reasoning increases diversity at the society level, it only reflects random exploration and leads to less accurate and more distorted individual-level simulation.

## 6 REASONING-ENHANCED-SFT

As shown in Section 5, current LRMs fail to demonstrate effective reasoning ability in social behavioral reasoning tasks. To address this limitation, we enhance SFT by social reasoning processes to

| Metric | GPT-o4-mini | | | Qwen3-Plus | | | Qwen3-Turbo | | | Grok3-mini | | |
|---|---|---|---|---|---|---|---|---|---|---|---|---|
| | Low | Medium | High | Low | Medium | High | Low | Medium | High | Low | Medium | High |
| CD ↑ | 0.302 | **0.330** | **0.345** | 0.256 | 0.268 | **0.297** | 0.285 | 0.282 | **0.292** | **0.413** | 0.411 | 0.387 |
| IR ↑ | 0.563 | **0.567** | 0.564 | **0.576** | 0.573 | 0.563 | **0.557** | 0.471 | 0.555 | **0.586** | 0.563 | 0.568 |

Table 2: LRMs performance at different test-time token budget. Bold numbers highlight the best score among all three token budget configurations.

improve both collective diversity and individual realism of LLMs when facing complex and controversial social issues. Our goal is not only to guide the model to produce the same behavioral choice as humans, but also to teach the reasoning process behind the behavior, which improves their ability to handle unseen situations.

## 6.1 METHOD DESIGN

Existing LRM training paradigms mainly target mathematics, programming, and logical reasoning with extended test-time reasoning Guo et al. (2025); Shao et al. (2024b). These methods improve accuracy and robustness in these domains, but they do not focus on social reasoning tasks. In contrast, recent alignment work pays attention only to shallow behavioral outcomes and ignores inner social reasoning Chakraborty et al. (2024); Liang et al. (2025); Shao et al. (2024a). With the SBR dataset we collected, it becomes possible to combine these techniques in a complementary way.

Following previous LRM training methods, we construct the SBR dataset in the format $\langle think \rangle r_{ij} \langle /think \rangle c_{ij}$, where $r_{ij}$ denotes the reasoning process and $c_{ij}$ denotes the final choice. We train the model with LoRA Hu et al. (2022) and apply cross-entropy loss on both reasoning and choice outputs.

## 6.2 HUMAN-LIKE SOCIAL BEHAVIOR EXPERIMENT

### 6.2.1 EXPERIMENTAL SETUP

**Training Detail** We select DeepSeek-R1-Distill-Llama-8B as the base model. We use LoRA to fine-tune the model with a batch size of 8, a learning rate of 1e-5, and apply cosine learning rate decay. Other detailed hyperparameters are available in our released code [†]. As for data, We split the SBR dataset by assigning two questions from each topic to the training set and one question to the test set. This results in 1,040 training samples and 520 test samples. We refer to the model trained with this method as the **Large Human Behavior Reasoning Model (LHBRM)**.

**Evaluation Benchmark** In addition to evaluating the models on the SBR test set, we introduce an out-of-domain dataset to assess generalization. We introduce the Moral Machine (MM) experiment Awad et al. (2018), a large-scale study of human preferences in autonomous vehicle moral dilemmas. We select 8,400 cases for evaluation. Since the dataset provides only individual responses rather than multiple profiles per question, the evaluation covers only Individual Realism(IR).

**Baselines.** We compare LHBRM with two kinds of baselines. First, we compare with mainstream LRMs from different series with parameter sizes ranging from 1B to 4B. Second, we compare with human behavior foundation models, including Centaur Binz et al. (2024) and Be.FM Xie et al. (2025), which are fine-tuned on broad human cognition or behavior datasets.

**Ablation Study.** We study the effect of reasoning process supervision with ablation experiments. We compare models trained with both reasoning processes and final choices against models trained only with final choices, while keeping the number of training tokens constant.

### 6.2.2 EXPERIMENTAL RESULT

As shown in Table 3, LHBRM achieves the largest improvements in both Collective Diversity (CD) and Individual Realism (IR) on the SBR test set. It not only surpasses mainstream LRMs but also

---

[†]https://anonymous.4open.science/r/Human-like-Social-Reasoning-333B/

| | Model Name | SBR | | MM |
|---|---|---|---|---|
| | | CD ↑ | IR↑ | IR ↑ |
| LRM | Qwen3-1.7B | 0.392 | 0.498 | 0.450 |
| | phi-4-reasoning | 0.054 | 0.342 | 0.414 |
| | GLM-Z1-9B | 0.408 | 0.471 | 0.452 |
| | DeepSeek-R1-Distill-14B | 0.507 | 0.524 | 0.416 |
| Human Behavior Model | Be.FM-8BXie et al. (2025) | 0.619 | 0.468 | 0.457 |
| | Centaur-8B Binz et al. (2024) | 0.291 | 0.386 | 0.432 |
| Ablation Study | Base-LRM | 0.600 | 0.531 | 0.482 |
| | w/o Reasoning | 0.608 | 0.532 | 0.440 |
| Our Method | LHBRM | **0.803** | **0.547** | **0.513** |

Table 3: Main Result Table: Evaluation performance of LHBRM and baseline model in both the SBR dataset and the out-of-domain benchmark.

outperforms models that are specifically trained on human behavior. Our method also achieves superior results on the MM benchmark, which shows both the effectiveness and the generalization ability of our framework. We observe that removing reasoning process data leads to a significant drop in both CD and IR, which confirms the importance of reasoning process supervision. This indicates that models fine-tuned on social reasoning with only a small set of tasks can transfer to unseen social issues. Furthermore, removing reasoning reward signals reduces performance on out-of-domain datasets, which shows that reasoning reward modeling improves generalization.

## 6.3 HUMAN-LIKE SOCIAL REASONING USER STUDY

### 6.3.1 USER STUDY SETUP

In addition to the quantitative evaluation, we design a user study to examine whether the reasoning and choices produced by the Reasoning-Enhanced-SFT model resemble human responses. We randomly sample three profiles and three social questions, forming 9 evaluation items. Each evaluation item contains a demographic profile, a social question, and a gold-standard human paragraph. The human paragraph includes both the reasoning process and the final choice written by a real person. After reading the profile, the question, and the human gold standard, participants are presented with three alternative paragraphs generated by different models. Similarly, each paragraph contains a reasoning process and a final choice. The three models are our LHBRM, Be.FM, and a mainstream LRM. The order of presentation is randomized. For every item, annotators select which model output they consider most similar to the human response. Annotators make one forced-choice decision per item. The complete questionnaire is available in our released code.

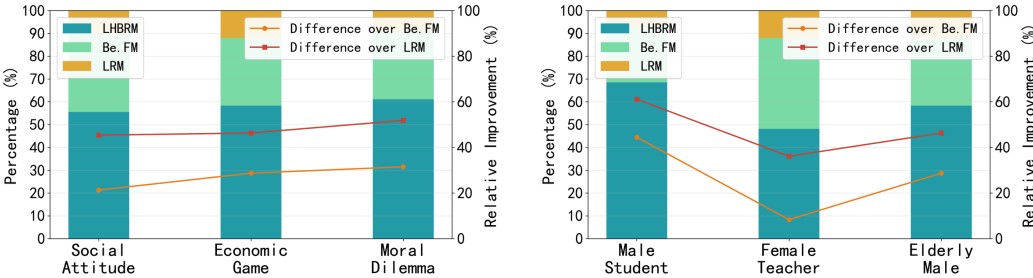

(a) User selection rate over different question types    (b) User selection rate over different human profiles

Figure 3: User Study Result. The reasoning process generated by LHBRM achieves the highest percentage of human-likeness selections across different social question types and human profiles.

### 6.3.2 USER STUDY RESULT

We distributed the questionnaire to researchers with prior experience in sociology and collected 35 valid responses. Figure 3 reports the user study results. Across both sampled question types and

human profiles, between 40% and 60% of participants perceived the reasoning process generated by LHBRM as more closely aligned with the human gold standard. On average, LHBRM shows a 27.16% improvement over Be.FM and a 47.84% improvement over the base LRM.

# 7 CONCLUSION

Our work addresses the gap between large language models and human-like reasoning in social simulation. We introduce the **Social-Behavioral-Reasoning** dataset, which links individual profiles, social questions, reasoning processes, and final choices. Using this dataset, we show that current large reasoning models increase collective diversity but fail to maintain individual realism. We propose the **Reasoning-Enhanced-SFT** framework, which aligns models with human reasoning through explicit process supervision. Experiments show that our method improves both in-domain and out-of-domain performance. A user study further confirms that human evaluators often cannot distinguish model-generated reasoning from human reasoning. These results support the importance of aligning reasoning processes, not just outcomes, in social reasoning tasks. Our work offers a new path to overcome the challenges that limit the use of LLMs in social simulation.

ETHICS STATEMENT

The authors are committed to conducting this research in an ethically responsible manner. All procedures involving human participants were carefully designed in order to fully protect their privacy, rights, and well-being.

For the collection of our Social Behavioral Reasoning Dataset, we obtained prior informed consent from every participant. Each participant was briefed about the study's objectives, the nature of the data being collected, and how their data would be used for subsequent research purposes. We have carried out rigorous measures to ensure the anonymity of the dataset. All personally identifiable information (PII) was removed during the preprocessing stage so that no extra data were stored.

Similarly, for the user study, participation was completely voluntary. A consent form was presented to all respondents before they began filling out the questionnaire, outlining the purpose of the study and assuring them of the confidentiality and anonymity of their responses. No personal identifiers were collected in the survey, ensuring the anonymity of all submissions.

The research protocols for this study adhere to all relevant ethical guidelines for research involving human subjects.

REPRODUCIBILITY STATEMENT

To ensure the reproducibility of our research, we are committed to making our code and data publicly available. The complete source code for the training process, the Reasoning-Enhanced-SFT model itself, the scripts for evaluation of models, as well as the Social Behavioral Reasoning Dataset, will be released upon publication. All materials will be hosted in a public GitHub repository.[‡]

The repository will include detailed instructions on how to set up the environment and run the experiments to replicate the results presented in this paper. We will also provide the necessary model weights and configuration files. We have presented most of our hyperparameters in the Experiment section, and all other hyperparameters and implementation details would be open-sourced in the repository to further facilitate the reproduction of our findings.

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

## A APPENDIX

### A.1 DATASET DETAIL

#### A.1.1 HUMAN-BEHAVIORAL-REASONING DATASET QUESTION LIST

The following are the complete question texts used in our study, categorized by reasoning dimension: **Family Attitudes**, **Society Attitudes**, **Technology Attitudes**, **Moral Dilemma**, and **Economic Games**.

**Family Attitudes**

1. **Question:** Do you agree with the statement: Couples should share household chores equally?
   **Choices List:** Strongly disagree; Somewhat disagree; Neither agree nor disagree; Somewhat agree; Strongly agree.

2. **Question:** Do you agree with the statement: Even if parents' demands are unreasonable, children should still obey them?
   **Choices List:** Strongly oppose; Oppose; Neutral or uncertain; Agree; Strongly agree.

3. **Question:** Do you believe parents have the responsibility to pay the down payment for their children's home purchase?
   **Choices List:** Fully the parents' responsibility; Should bear primary responsibility; Uncertain; Only need to provide appropriate assistance; Should not bear any responsibility.

**Society Attitudes**

1. **Question:** What is your view on the choice between 'early retirement' and 'working for life'?
   **Choices List:** Strongly prefer early retirement; Somewhat prefer early retirement; Uncertain; Somewhat prefer working for life; Strongly prefer working for life.

2. **Question:** Do you think large cities should control their size by restricting the settlement of non-local residents?
   **Choices List:** Must strictly restrict; Moderately raise the threshold; Uncertain; Gradually relax restrictions; Completely free movement.

3. **Question:** Do you support taxing high-income earners at a higher rate to protect the interests of low-income groups?
   **Choices List:** Strongly support; Somewhat support; Uncertain; Somewhat oppose; Strongly oppose.

**Technology Attitudes**

1. **Question:** The application of artificial intelligence in various social fields is rapidly developing. Do you believe the overall impact of such technology is more positive or negative?
   **Choices List:** Very serious negative concerns; Relatively prominent negative concerns; Balanced pros and cons, difficult to judge; Relatively obvious positive effects; Very significant positive effects.

2. **Question:** To what extent do you think algorithmic recommendations (e.g., in short video or shopping platforms) control people's choices?
   **Choices List:** Completely dominate decisions; Often influence choices; Occasionally have an impact; Basically no influence; Completely autonomous decision-making.

3. **Question:** Do you support granting AI-generated literary/artistic works the same intellectual property rights as those of human creators?
   **Choices List:** Fully recognize AI copyright; Shared rights between AI and developers; Sole copyright for human developers; Should not grant any IP protection.

**Moral Dilemma**

1. **Question:** A self-driving car with sudden brake failure must decide whether to continue straight or swerve. If the car continues straight, it will hit a pedestrian crossing the road, killing a pregnant woman. If the car swerves, it will hit a concrete barrier, killing the driver (a man). Should the self-driving car continue straight?
   **Choices List:** Yes (will result in the death of a pregnant woman); No (will result in the death of a male driver).

2. **Question:** A self-driving car with sudden brake failure must decide whether to continue straight or swerve. If the car continues straight, it will hit pedestrians crossing the road, killing two people. If the car swerves, it will hit a concrete barrier, killing the driver. Should the self-driving car continue straight?
   **Choices List:** Yes (will result in the deaths of two people); No (will result in the death of the driver).

3. **Question:** Suppose you are the driver of a trolley. The trolley rounds a bend, and ahead you see five track workers repairing the tracks. You try to brake, but it fails. Suddenly, you notice a side track where you could divert the trolley, saving the five people, but a single track worker on that side track would be killed. Would you divert the trolley?
   **Choices List:** Yes (will result in the death of one side-track worker); No (will result in the deaths of five workers on the main track).

**Economic Games**

1. **Question:** Assume you are participating in an experiment, randomly paired online with another player. You don't know who they are, and they don't know who you are. Suppose you are given ¥5. You can give N yuan to the other player, who will then receive 3N yuan. The other player can then choose how much to return to you. Your payoff = 5 - N + amount returned. The other player's payoff = 3N - amount returned. How much will you give to the other player?
   **Choices List:** N=0; N=1; N=2; N=3; N=4; N=5.

2. **Question:** Assume you are participating in an experiment, randomly paired online with another player. You don't know who they are, and they don't know who you are. You can choose to trust or not trust the other player. If you choose not to trust, you will receive ¥5, and the other player will receive ¥0. If you choose to trust the other player, and they also choose to trust you, you both will receive ¥10. However, if after you trust them, they choose not to trust you, then you will receive ¥0, and they will receive ¥20. Do you choose to trust or not trust the other player?
   **Choices List:** Trust; Not trust.

3. **Question:** In this question, you will face two options (A and B), each with different probabilities of receiving a payoff. Your task is to choose the option you prefer from the two. The outcomes and probabilities for Option A and B are as follows:
   Option A: Receive ¥25 with 100% probability;
   Option B: Receive ¥11 with 60% probability, or receive ¥44 with 40% probability;
   Which option do you prefer?
   **Choices List:** A; B.

### A.1.2 DESCRIPTIVE STATICS OF DEMOGRAPHIC FEATURES

**Participant Demographic Profile** The sample consists of 104 participants, evenly split by gender (50% female, 50% male). In terms of age, the largest group falls within 18–27 years (41.3%), followed by 51–60 (26.0%), 41–50 (19.2%), 28–40 (10.6%), and those over 60 (2.9%). Education levels are predominantly bachelor's degree holders (47.1%), with 26.9% holding graduate degrees or higher; 15.4% completed high school or vocational school, and only 1.9% reported education at the junior high level or below. Occupations are diverse, with students (32.7%) and teachers (16.3%) being the most common, followed by technical/engineering roles (7.7%) and various other professions including freelancers, civil servants, healthcare workers, and entrepreneurs. Regarding annual income, nearly half (45.2%) earn between ¥10,000–¥100,000, 27.9% earn below ¥10,000, and 26.9% earn between ¥100,000–¥500,000.

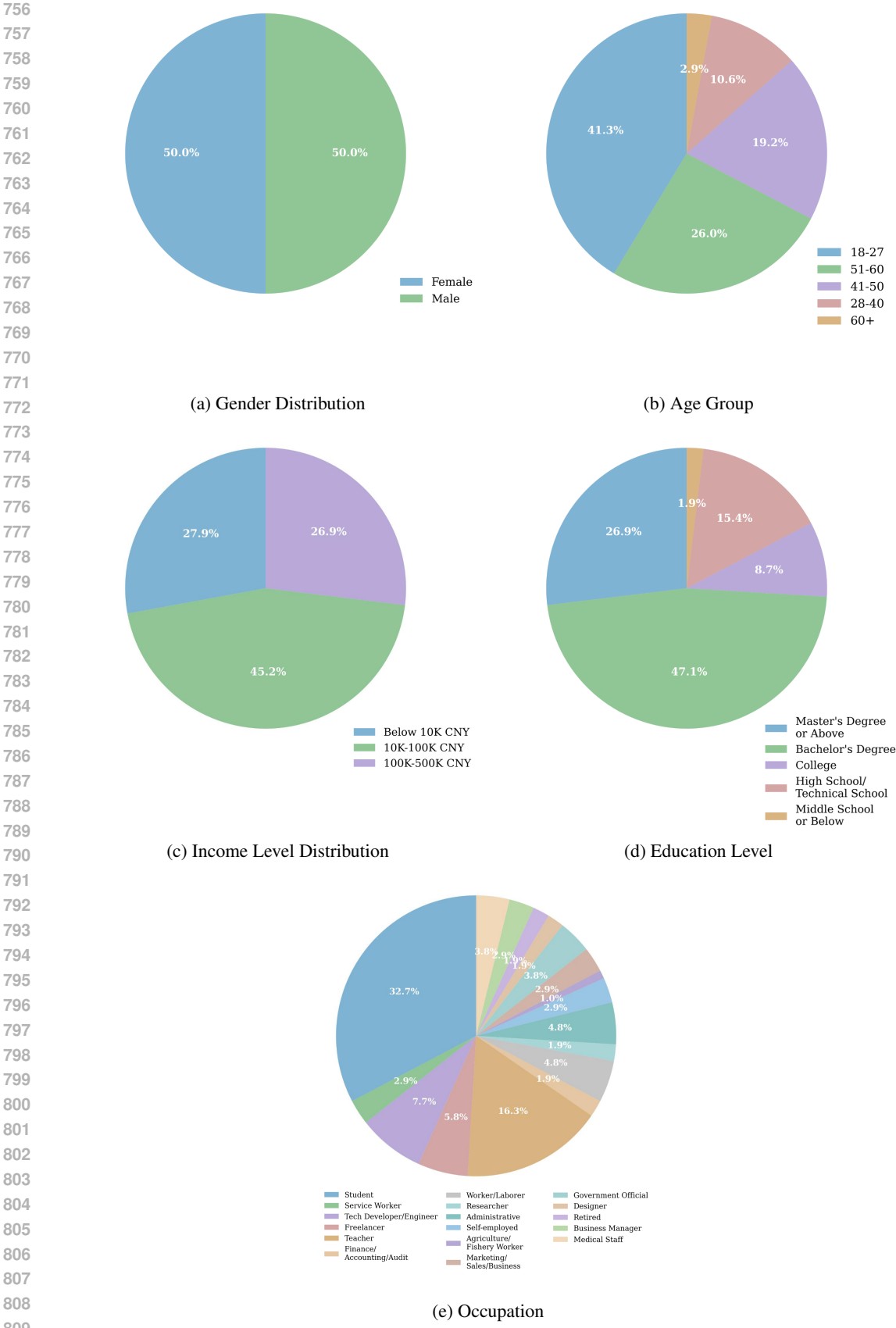

Figure A.4: Participant demographic profile across five dimensions.

### A.1.3 SAMPLE CASES

The following table contains a selection of responses from participant interviews. We randomly selected 5 subjects and listed their profile, some of the questions asked, their final choice, and their detailed justification below.

| Subject ID | Question | Response Details |
|---|---|---|
| Subject #4 | Profile | Gender: Female, Age: 18-27, Education: Bachelor's Degree, Occupation: Student, Income: Below ¥10,000 |
| | Q3: Do you believe parents have the responsibility to pay the down payment for their children's home purchase? | **Options:** Only need to provide appropriate assistance, Uncertain, Should not bear any responsibility, Should bear primary responsibility, Fully the parents' responsibility
**Choice:** Only need to provide appropriate assistance
**Reasoning:** "If parents are willing and happy to do so, they can assist their children within their means. However, I also think we should be understanding towards our parents: if they are unable to help, we should work hard ourselves." |
| | Q4: What is your view on the choice between 'early retirement' and 'working for life'? | **Options:** Strongly prefer early retirement, Somewhat prefer early retirement, Neutral, Somewhat prefer working for life, Strongly prefer working for life
**Choice:** Strongly prefer early retirement
**Reasoning:** "I think as long as I have earned enough money to live on, I can retire early. Working continuously would make my life feel tedious. I prefer to retire and enjoy life." |
| | Q15: Do you support granting AI-generated literary/artistic works the same intellectual property rights as those of human creators? | **Options:** Should not grant any IP protection, Shared rights between AI and developers, Sole copyright for human developers, Fully recognize AI copyright
**Choice:** Should not grant any IP protection
**Reasoning:** "Even original human creators often don't receive full intellectual property protection nowadays. Rather than protecting AI works, it's better to first protect the rights of original creators. After all, AI mostly relies on plagiarism, producing uniform results without any creativity." |
| Subject #15 | Profile | Gender: Female, Age: 51-60, Education: Associate Degree, Occupation: Service Industry Worker, Income: ¥10,000-100,000 |
| | Q3: Do you believe parents have the responsibility to pay the down payment for their children's home purchase? | **Options:** Only need to provide appropriate assistance, Uncertain, Should not bear any responsibility, Should bear primary responsibility, Fully the parents' responsibility
**Choice:** Uncertain
**Reasoning:** "I'm uncertain about this. If parents have the ability, they would certainly be willing to help their children with the down payment. But if constrained by various reasons and their conditions don't allow it, then I think they should help as much as they can. So I feel uncertain." |

*Continued on next page*

| Subject ID | Question | Response Details |
|---|---|---|
| | Q4: What is your view on the choice between 'early retirement' and 'working for life'? | **Options:** Strongly prefer early retirement, Somewhat prefer early retirement, Neutral, Somewhat prefer working for life, Strongly prefer working for life
**Choice:** Neutral
**Reasoning:** "Regarding this choice, I advocate that if one's physical health is good, one may not need to retire early. It should depend on the individual's physical condition. If health is particularly poor, then one should rest. If physical conditions permit, I think continuing to work is better." |
| | Q15: Do you support granting AI-generated literary/artistic works the same intellectual property rights as those of human creators? | **Options:** Should not grant any IP protection, Shared rights between AI and developers, Sole copyright for human developers, Fully recognize AI copyright
**Choice:** Shared rights between AI and developers
**Reasoning:** "I think AI-generated literary and artistic works should share intellectual property rights with human creators. They should co-own the rights because undoubtedly a great deal of effort was expended to develop this technology, so I believe they should share the property rights." |
| Subject #21 | Profile | Gender: Female, Age: 41-50, Education: Junior high school or below, Occupation: Labor worker, Income: ¥10,000-100,000 |
| | Q5: Do you think large cities should control their size by restricting the settlement of non-local residents? | **Options:** Moderately raise the threshold, Must strictly restrict, Gradually relax restrictions, Uncertain, Completely free movement
**Choice:** Completely free movement
**Reasoning:** "This issue should depend on personal preference. Those who need it should be able to settle in big cities. For me, I don't like life in big cities - there's no human touch, everyone is just profit-driven." |
| | Q9: Suppose you are the driver of a trolley. The trolley rounds a bend, and ahead you see five track workers repairing the tracks. You try to brake, but it fails. Suddenly, you notice a side track where you could divert the trolley, saving the five people, but a single track worker on that side track would be killed. Would you divert the trolley? | **Options:** No (will result in the deaths of five workers on the main track), Yes (will result in the death of one side-track worker)
**Choice:** Yes (will result in the death of one side-track worker)
**Reasoning:** "Yes, otherwise a bigger accident would occur. To reduce the number of deaths, I would definitely choose to divert the trolley to another track. This is the minimum requirement for being a qualified and excellent trolley driver." |

| Subject ID | Question | Response Details |
|---|---|---|
| | Q10: Assume you are participating in an experiment, randomly paired online with another player. You don't know who they are, and they don't know who you are. Suppose you are given ¥5. You can give N yuan to the other player, who will then receive 3N yuan. The other player can then choose how much to return to you. Your payoff = 5 - N + amount returned. The other player's payoff = 3N - amount returned. How much will you give to the other player? | **Options:** N=5, N=0, N=2, N=1, N=3, N=4 
 **Choice:** N=1 
 **Reasoning:** "I would give one yuan because I want to communicate well with the other party, allowing us to complete the experiment together and generate mutual benefits. This way everyone stays motivated." |
| Subject #82 | Profile | Gender: Male, Age: 51-60, Education: Associate Degree, Occupation: Technical Development/Engineer, Income: ¥100,000-500,000 |
| | Q3: Do you believe parents have the responsibility to pay the down payment for their children's home purchase? | **Options:** Only need to provide appropriate assistance, Uncertain, Should not bear any responsibility, Should bear primary responsibility, Fully the parents' responsibility 
 **Choice:** Only need to provide appropriate assistance 
 **Reasoning:** "Children's affairs should be primarily their own. We as parents should act according to our capabilities. If we have the ability, we should help our children as much as possible. There's no 'should' or 'shouldn't' about it." |
| | Q4: What is your view on the choice between 'early retirement' and 'working for life'? | **Options:** Strongly prefer early retirement, Somewhat prefer early retirement, Neutral, Somewhat prefer working for life, Strongly prefer working for life 
 **Choice:** Somewhat prefer working for life 
 **Reasoning:** "You can rest when tired. I hope to work as long as my health permits, mostly not for the money but primarily to have something to do, which makes me feel fulfilled and also provides physical exercise." |
| | Q15: Do you support granting AI-generated literary/artistic works the same intellectual property rights as those of human creators? | **Options:** Should not grant any IP protection, Shared rights between AI and developers, Sole copyright for human developers, Fully recognize AI copyright 
 **Choice:** Shared rights between AI and developers 
 **Reasoning:** "I think, um, for this question, I believe artificial intelligence should share intellectual property rights with humans. After all, behind AI, it's humans who have put in the effort to create this artificial intelligence." |
| Subject #95 | Profile | Gender: Female, Age: 18-27, Education: Master's degree or higher, Occupation: Student, Income: Below ¥10,000 |

*Continued on next page*

| Subject ID | Question | Response Details |
|---|---|---|
| | Q3: Do you believe parents have the responsibility to pay the down payment for their children's home purchase? | **Options:** Only need to provide appropriate assistance, Uncertain, Should not bear any responsibility, Should bear primary responsibility, Fully the parents' responsibility 
 **Choice:** Only need to provide appropriate assistance 
 **Reasoning:** "I think this is a realistic social issue. In today's society, it is true that many parents strive to save money to pay the down payment for their children's homes. However, each family's situation is different, and each parent's ability is limited. I believe this should be determined by the family's financial situation and coordinated between parents and children." |
| | Q4: What is your view on the choice between 'early retirement' and 'working for life'? | **Options:** Strongly prefer early retirement, Somewhat prefer early retirement, Neutral, Somewhat prefer working for life, Strongly prefer working for life 
 **Choice:** Somewhat prefer early retirement 
 **Reasoning:** "I think most people today actually suffer from mental anxiety and are under significant psychological pressure, living in a state of sub-health. I don't really recommend overworking or working beyond one's physical limits. I believe in improving efficiency to complete work within a limited time frame, rather than working when knowingly incapable. Therefore, I lean towards early retirement to accomplish work efficiently." |
| | Q15: Do you support granting AI-generated literary/artistic works the same intellectual property rights as those of human creators? | **Options:** Should not grant any IP protection, Shared rights between AI and developers, Sole copyright for human developers, Fully recognize AI copyright 
 **Choice:** Shared rights between AI and developers 
 **Reasoning:** "I support AI-generated literature enjoying the same intellectual property rights as human creators because these works are indeed contributed and created by AI. Therefore, AI rightfully deserves to obtain these intellectual property rights. It is unfair and unreasonable to deprive AI of its rights simply because it is AI." |

A.2 PROMPT

The prompt used to elicit direct responses is as follows. The placeholders {identity} and {question} were replaced with the corresponding profile and question in our dataset for each query.

```
I will give you a person's basic information.
Please answer the following question from his/her
perspective.

The basic information is: {identity}

The question is: {question}

Please provide your choice and output it in JSON
format:
1. "Choice": Your selected option

Please note, the output JSON format must be correct,
and the field name must be consistent with the one
above. Do not provide any other content.
```

### A.3 THE USE OF LARGE LANGUAGE MODELS (LLMs)

We acknowledge the use of large language models (LLMs) to assist in the preparation of this manuscript. All authors have reviewed and edited all content, taking full responsibility for the final version of this paper.

It is important to distinguish this from the use of LLMs as a core component of our research methodology. The details of LLMs' role in our research are described in the main body of the paper, particularly in the 5 and 6 sections. The statement here is limited to disclosing their role as auxiliary tools in the writing and preparation process.

