# OpenReview forum: "Eliciting Human-like Social Reasoning in Large Language Models"
_ICLR.cc/2026/Conference — ICLR 2026 Conference Withdrawn Submission_

### Official Review · Reviewer_TdxY · 2025-10-18

**Soundness:** 3
**Presentation:** 3
**Contribution:** 2
**Rating:** 2
**Confidence:** 3

**Summary:**

This paper introduces the Social-Behavioral-Reasoning (SBR) dataset and a Reasoning-Enhanced Supervised Fine-Tuning (SFT) framework that jointly trains models on both reasoning processes and final decisions, improving the human-likeness and interpretability of large language models in social reasoning tasks.

**Strengths:**

1. Clearly identifies a meaningful gap, the lack of social reasoning capability in current LLMs, and frames it as a concrete research problem.

2. Proposes a straightforward yet effective Reasoning-Enhanced SFT approach that aligns both reasoning processes and behavioral outputs.

3. Constructs and releases the SBR dataset that explicitly links human profiles, social questions, reasoning texts, and behavioral choices, providing a valuable foundation for future studies.

**Weaknesses:**

1. The approach mainly involves building a dataset and applying standard SFT/LoRA fine-tuning; the improvements largely stem from data rather than algorithmic innovation.

2. Experiments are confined to quantitative comparisons on Collective Diversity (CD) and Individual Realism (IR), without richer qualitative or interpretive insights.

3. While well-structured, the dataset is relatively small (1,560 samples) and lacks systematic comparison with prior social or moral reasoning datasets.

**Questions:**

How would you utilize the remaining space for more analysis with depth?

---

### Official Review · Reviewer_foVw · 2025-10-31

**Soundness:** 2
**Presentation:** 2
**Contribution:** 2
**Rating:** 4
**Confidence:** 2

**Summary:**

This paper introduces the Social-Behavioral-Reasoning (SBR) dataset and a fine-tuning method called Reasoning-Enhanced-SFT, aiming to improve LLMs’ ability to perform human-like social reasoning. The authors analyze reasoning models and find that longer reasoning increases collective diversity but decreases individual realism. The proposed method jointly aligns reasoning traces and behavioral outcomes, leading to improvements on the SBR dataset and limited out-of-domain generalization.

**Strengths:**

1.  Provides a novel dataset linking human profiles, reasoning processes, and behavioral choices.

2. Offers empirical insight into the diversity–realism trade-off in reasoning models.

**Weaknesses:**

1. Lack of novelty. The proposed Reasoning-Enhanced-SFT is a straightforward extension of existing SFT with reasoning supervision, and the improvements may stem mainly from additional data rather than methodological innovation.

2. Limited contribution. The SBR dataset is small and simple, which restricts both the generality and the depth of the findings. The analysis remains mostly descriptive without deeper insight.

3. The claimed gains in diversity and realism are largely in-domain, and the external evaluation is narrow in scope.

**Questions:**

1. How was the reasoning quality and consistency in SBR verified?

2. How would this approach generalize to other domains of social reasoning?

3. Could the observed gains be explained by longer reasoning sequences rather than true alignment with human reasoning?

---

### Official Review · Reviewer_gLzv · 2025-11-01

**Soundness:** 1
**Presentation:** 2
**Contribution:** 1
**Rating:** 2
**Confidence:** 4

**Summary:**

This paper investigates whether large reasoning models (LRMs) can exhibit human-like social behavioral reasoning. The authors first construct a small Social-Behavioral-Reasoning (SBR) dataset consisting of demographic profiles, social questions, free-form human rationales, and final choices. They then evaluate a set of existing LRMs and observe that test-time reasoning tends to increase “collective diversity” while reducing “individual realism.” To address this, the paper proposes a “Reasoning-Enhanced SFT” approach that trains models on both human rationales and decisions. Experiments on the SBR benchmark and a moral-dilemma dataset, together with a small user study, suggest modest improvements in perceived human-likeness of reasoning.

**Strengths:**

Clear motivation: The paper highlights an important challenge of simulating human social behavior and reasoning with LLMs, a topic of growing interest in both machine learning and computational social science.

Data collection effort: Constructing a dataset with paired profiles, open-ended rationales, and decisions reflects a meaningful annotation effort.

Problem space relevance: The work touches on value alignment, human-agent interaction, and social simulation — areas with potential downstream societal impact.

**Weaknesses:**

1. Severe Limitations in Problem Formulation and Scope
While the paper aims to study "social reasoning," its operationalization is overly simplistic. The task is confined to answering single-turn, context-free survey questions, which is more akin to preference elicitation than complex social reasoning. The chosen questions (e.g., on household chores or early retirement) lack the elements of strategic interaction, dynamic belief updating, or rich social context that are central to genuine social cognition. Consequently, the problem formulation is not sufficiently complex to support broad claims about eliciting "human-like social reasoning" in LLMs.

2. Methodological Simplicity Compounded by Critical Data Scarcity
The core methodological proposal, "Reasoning-Enhanced-SFT," is a standard application of supervised fine-tuning on rationale + answer pairs. While methodologically simple, its primary weakness lies in the dataset's extremely small scale. Fine-tuning on merely 1,560 training examples from 104 individuals is highly unlikely to impart generalizable social reasoning abilities. Instead, it is far more plausible that the model is overfitting to the specific linguistic styles and response patterns present in this small sample. The reported improvements, including the out-of-domain results, may stem from learning superficial stylistic cues rather than genuine cognitive processes. The authors do not provide sufficient evidence (e.g., rigorous analysis of failure cases or generalization across more diverse tasks) to disentangle true reasoning from stylistic imitation.

3. Inability to Validate Key Claims Due to Data Scale and Accessibility
The paper's central claims—particularly the "diversity-realism trade-off" and the effectiveness of the proposed solution—rest entirely on a dataset that is too small to yield statistically meaningful conclusions. Social attitudes and reasoning are incredibly diverse; findings from 104 participants cannot be reliably generalized. Metrics like "Individual Realism" lose their meaning when the ground truth for a given demographic profile is based on a single data point. This small sample size makes the entire quantitative evaluation fragile and potentially misleading.

Furthermore, the lack of access to the full human rationales during the review process prevents an independent assessment of their quality and complexity, which is crucial for a dataset-centric contribution. Without this, it is impossible to verify whether the human data provides a sufficiently strong and nuanced signal for training.

**Questions:**

Your central claim is that Reasoning-Enhanced-SFT improves social reasoning. However, given the extremely small dataset (104 participants), a more parsimonious explanation is that the model is overfitting to the specific phrasing and stylistic patterns of this sample. How can you definitively distinguish genuine improvement in social cognition from mere stylistic imitation? For instance, did you perform any qualitative analysis on the generated reasoning to show it goes beyond surface-level mimicry? What steps were taken to validate that the out-of-domain performance is not also an artifact of stylistic generalization?

---

### Official Review · Reviewer_wp3H · 2025-11-02

**Soundness:** 2
**Presentation:** 3
**Contribution:** 2
**Rating:** 4
**Confidence:** 3

**Summary:**

The paper asks whether large “reasoning” LLMs (LRMs) actually think like humans in social decision-making. It builds a new Social-Behavioral-Reasoning (SBR) dataset pairing demographic personas, socially contested questions, human written reasoning, and final choices. Two complementary metrics are introduced: Collective Diversity (distributional entropy over choices across personas) and Individual Realism (closeness to a specific persona’s choice). Across many LRM–baseline pairs, LRMs raise diversity but often degrade persona-level realism, and granting more test-time reasoning tokens amplifies this divergence—suggesting added “thinking” increases stochastic branching rather than faithful persona modeling. To address this, the authors propose Reasoning-Enhanced SFT that supervises both human rationales and outcomes. The resulting model (LHBRM) improves both diversity and realism, shows out-of-domain gains on Moral Machine, and is preferred by human raters for realism of reasoning.

**Strengths:**

1 Problem framing + evaluation lens are spot-on.
The diversity–realism two-axis evaluation directly captures the core tension in using LLMs as human proxies (population heterogeneity vs. persona fidelity), moving beyond one-number “accuracy.”

2 Actionable negative finding that challenges assumptions.
The paper shows that giving LRMs more test-time reasoning increases distributional diversity but hurts persona-level realism—an impactful, counter-intuitive result that reframes how “thinking tokens” should be interpreted in social reasoning.

3 Simple, generalizable training recipe with measurable gains.
Reasoning-Enhanced SFT (supervising human rationales + outcomes) is architecture-agnostic and empirically improves both axes, making it easy for the community to adopt and build upon.

**Weaknesses:**

1 Dataset scope and representativeness are too narrow.
The SBR corpus has ~hundreds of subjects and 15 scenarios, with limited demographic, cultural, and ideological coverage. Online-recruitment selection effects are not quantified, weakening external validity.

2 Construct validity of the metrics is questionable.
The diversity metric (entropy) can reward stochastic noise rather than faithful recovery of group structure; the realism metric assumes an ordinal-to-interval mapping on [0,1] that may distort semantic distances. Sensitivity to binning, option count, and mapping choices is not reported.

3 The mechanism behind “more reasoning tokens ↓ realism” is under-substantiated.
The paper attributes the effect to stochastic branching, but does not adequately control for temperature/top-p, sampling count, deterministic decoding, reflect-and-revise loops, or chain-length constraints; thus causal interpretation remains weak.

4 Persona prompting is fragile and may leak stereotypes.
Prompt template, attribute order/density, and conflicting attributes likely drive outputs. There is no systematic robustness analysis (prompt variants, attribute permutations) nor checks for prompt-induced stereotype leakage.

5 Comparison fairness and supervision budget confounds.
LRMs and baselines are not clearly matched for compute (context window, token budget, generation time). Reasoning-Enhanced-SFT supervises both process and outcome, potentially increasing total supervised tokens vs. “outcome-only” baselines, confounding attribution of gains.

6 Process supervision may learn post-hoc rationalizations and amplify bias.
Human “reasons” can be justificatory rather than causal; the model may imitate stylistic rationales, not decision mechanisms. Safety auditing for harmful or discriminatory patterns in learned rationales is limited.

7 Evaluation breadth and rigor are insufficient.
OOD testing is limited (Moral Machine, realism only); human preference studies are small-N with sparse statistical reporting (CIs, multiple-comparison control, inter-rater reliability). Error analysis by persona/issue is thin, and reproducibility details (code, seeds, exact configs) are incomplete.

**Questions:**

The same as the weaknesses

---

### Note · Authors · 2025-12-09

I have read and agree with the venue's withdrawal policy on behalf of myself and my co-authors.